# Prostate Apoptotic Induction and NFκB Suppression by Dammarolic Acid: Mechanistic Insight into Onco-Therapeutic Action of an Aglycone Asiaticoside

**Ahmed Alafnan** [1] , **Talib Hussain** [1,*], **Syed Mohd Danish Rizvi** [2,*] , **Afrasim Moin** [2] **and Abdulwahab Alamri** [1]

1  Department of Pharmacology and Toxicology, College of Pharmacy, University of Hail, Hail P.O. Box 2240, Saudi Arabia; a.alafnan@uoh.edu.sa (A.A.); a.alamry@uoh.edu.sa (A.A.)
2  Department of Pharmaceutics, College of Pharmacy, University of Hail, Hail P.O. Box 2240, Saudi Arabia; a.moinuddin@uoh.edu.sa
*  Correspondence: mdth_ah@yahoo.com (T.H.); sm.danish@uoh.edu.sa (S.M.D.R.)

**Abstract:** Prostate cancer (PCa) is addressed as the second most common form of onco-threat worldwide and is usually considered as the major cause of mortality in men. Recent times have seen a surge in exploration of plant-derived components for alternative therapeutical interventions against different oncological malignancies. Dammarolic acid or Asiatic acid (AsA) is an aglycone asiaticoside that has been reported for its efficacy in several ailments including cancer. The current study aimed to investigate the anti-proliferative potency of AsA against human prostate cancer PC-3 cells. Purified AsA was diluted and PC-3 cells were exposed to 20, 40, and 80 μM concentration and incubated for 24 h. Post-exposure, PC-3 cells showcased a substantial loss of their viability at 20 μM ($p < 0.05$), moreover, this reduction in cell viability escalated proportionally with an increase in AsA at concentrations of 40 and 80 μM ($p < 0.01$; $p < 0.001$) respectively. AsA-impelled loss of cellular viability was also evident from the acridine orange-stained photomicrographs, which was also used to quantify the viable and apoptotic cells using Image J software. Additionally, quantification of ROS within PC-3 cells also exhibited an increase in DCF-DA-mediated fluorescence intensity post-exposure to AsA in a dose-dependent manner. AsA-induced apoptosis in PC-3 cells was shown to be associated with augmented activity of caspase-3 proportionally to the AsA concentrations. Thus, initially, this exploratory study explicated that AsA treatment leads to anti-proliferative effects in PC-3 cells by enhancing oxidative stress and inciting apoptosis en route to onset of nuclear fragmentation.

**Keywords:** prostate cancer; PC-3 cell; Asiatic acid; apoptosis; anti-proliferative

## 1. Introduction

Prostate cancer or PCa represent a common malignant tumor occurring within the genitourinary system in males and was accountable for 1,414,259 or 7.3% of new cancer-related cases during 2020 [1]. PCa is also known to be the foremost reason behind cancer-related mortalities among males globally. It has been reported that males having African ancestry are correlated with higher chances of developing the disease more aggressively than other ethnic groups, making them vulnerable to high mortality rates [2,3]. Furthermore, the highest amount of new PCa cases, accounting for 33.5%, were recorded alone in Europe followed by Asia (26.2%). Caribbean accounts for the highest amount of cancer-related deaths after the USA [1]. This debilitating oncological malignancy is associated with a number of varying risk factors among which the prominent ones include apoptosis modulation, genetically and epigenetically governed factors involved in the onset, proliferation, and metastasis [4].

Standard therapeutical management of prostate-related oncological malignancies relies upon surgical, chemo/radio-therapeutics, and hormonal or immunotherapeutical interventions. However, these interventions exert their effects synergistically where the

surgical removal of tumor growth is followed by either radio or chemotherapy or a combination of both. Chemotherapeutics involved in management of these malignancies are documented for their cytotoxic effects en route to instigation of apoptosis and are also known to cause arrest of tumor cells in different phases of cell cycle. The current treatment regime for hormone refractory PCa (HRPC) is docetaxel (DTX), which exerts its effect by destabilizing the microtubules homeostasis within the cells [5]. However, prolonged medication with docetaxel is reported to induce drug-specific resistance within individuals suffering from PCa, resulting primarily from mutations within microtubules with concomitant pump-mediated efflux of drug [6,7].

Nevertheless, the chemotherapeutic-based management of prostate cancer patients is also followed by destructive side-effects, which commonly have catastrophic effects on bone-marrow-derived blood cells, hair follicles, and cells within the oral cavity and digestive tract [7].

In overcoming these debilitating effects, herbal medicines constituted by phytoactive constituents are postulated to be indispensable and also supposed to cause substantial reduction in cost associated with chemotherapeutics [8]. Herbal medicines represent approximately 60% of all the chemotherapeutics that are currently being explored in developing novel anti-cancer drugs [9,10]. One such important bioactive compound is Asiatic acid, which belongs to the family of triterpenoids that have received serious consideration over the past several years. Dammarolic acid or Asiatic acid (AsA) represents pentacyclic triterpenoids, substituted by ursane at 28 positions, and -OH group at 2, 3, and 23 positions. AsA is a major constituent of *Centella asiatica*, *Purnella vulgaris*, *Nepeta hindostana*, *Eucalyptus perriniana,* and *Psidium guajava*. [11]. Earlier it was elucidated that AsA possess intrinsic characteristics that allow it to exert opposing effects on inflammation, diabetes, and cancer [12]. Anti-cancer efficacy of AsA was earlier reported in vitro in both the hepatic and breast cancer cells and was recently deduced to be linked with instigation of apoptosis. Moreover, AsA is further believed to negatively regulate the expression level of NF-κB, p38, MAPK, and ERKs, along with Bcl-2 and caspase proteins [13,14].

NF-κB is an important transcription factor playing prominent roles during inflammation and carcinogenesis and, therefore, is regarded as a plausible novel target for therapeutical intervention against cancer [15–17]. Chronic activation of NF-κB in cancer assists in the onset; progression of tumor; metastasis; and resistance towards chemotherapeutics by augmenting the expression of various growth factors, pro-inflammatory cytokines, chemokines, and anti-apoptotic factors [17–20]. To the best of our knowledge, there appears to be a paucity of scientific literature exploring the potency of AsA in modulating NF-κB expression within human prostate cancer PC-3 cells. Therefore, this preliminary investigation tries to revisit the anti-proliferative characteristics of AsA on PC-3 cells. It was hypothesized during this investigation that exposure of PC-3 cells with AsA would play a substantial role in restraining the proliferation of human-derived androgen-independent PC-3 cells by escalating ROS, which would further instigate apoptosis via modulation of caspase activity and NF-κB expression.

## 2. Materials and Methods

### 2.1. Materials

Asiatic acid (AsA), 3-(4,5-dimethylthiazol-2-yl)-2,5-diphenyl tetrazolium bromide (MTT), *N*-Acetyl cysteine (NAC), 2,7-dichlorodihydrofluorescein diacetate (DCFH-DA), Acridine orange, ethidium bromide (EtBr) solution, Trypan blue, and capase-3 inhibitors (Z-DEVD-FMK) were obtained from Sigma, St. Louis, MO, USA. Roswell Park Memorial Institute (RPMI 1640) and antibiotic-antimycotic solution were purchased from Himedia, Pune, India, whereas fetal bovine serum (FBS) used during in vitro culture was obtained from Gibco, West Chester, PA, USA. All the real-time primers employed in the study were synthesized and procured from Integrated DNA Technologies (IDT), Coralville, IA, USA. Verso cDNA synthesis kit and DyNAmoColorFlash SYBR Green qPCR Kit were obtained from Thermo-Scientific, Waltham, MA, USA.

*2.2. Methods*

2.2.1. Cell Culture and Its Maintenance

Androgen-independent human prostate cancer PC-3 cells were obtained from the cell repository of National Centre for Cell Sciences (NCCS), Pune, India. The PC-3 cells were maintained in Roswell Park Memorial Institute (RPMI 1640) during the entire course of experimentation. The mediums were also supplemented with 10% FBS (*v/v*) and 1% antibiotic-antimycotic solution (*v/v*). Cells were incubated under optimum cell culture conditions constituted by 5% $CO_2$ at 37 °C. Cells were routinely observed and passaged after attaining ~90% confluency within T-25 flask. Cells were used in pre-determined numbers in accordance with the assay after ascertaining the count of live cells through 0.4% trypan blue dye solution aided by hemocytometer, Sigma, St. Louis, MO, USA.

2.2.2. Assessment of AsA Mediated Toxicity

To explore AsA-induced cytotoxicity (if any) on PC-3 cells, tetrazolium-based MTT assay was undertaken as described previously [21]. PC-3 cells were seeded at a density of $5 \times 10^3$ cell/well in a 96-well format and allowed to adhere under optimum culture conditions as stated. After adherence, the cells were exposed to varying concentrations of AsA viz. 20, 40, and 80 μM, and were again subjected to optimum culture conditions for 24 h. Post-incubation, media containing AsA in each well were exchanged with 10 μL of MTT dye (5 mg/mL) and the plate was left for another incubation of 4 h under standard culture conditions. Thereafter, 100 μL of DMSO was included in each well and the plate was vortexed gently at room temperature for nearly 30 min. Finally, the absorbance of solubilized formazan crystals was recorded at 490 nm using a spectrophotometer (Bio-Rad, Hercules, CA, USA). The cytotoxicity of AsA PC-3 cell line was annotated as cell viability percent (%) in comparison with untreated control cells and estimated as $A_{test} \times 100/A_{control}$, where $A_{test}$ = absorbance of treated groups and $A_{control}$ = absorbance of control.

2.2.3. Evaluation of AsA Effects on Alteration of ROS Levels within PC-3 Cells

ROS generation within human prostate cancer PC-3 cells after treatment with AsA was analyzed using quantitative method based on DCFH-DA-mediated fluorescence intensity as described earlier [22]. Initially, $2 \times 10^4$ C-3 cells/well were placed in a black bottom 96-well format and left overnight to facilitate their adherence under culture conditions. Post adherence, PC-3 cells were exposed to varying concentrations of AsA (20–80 μM) and were incubated for the next 12 h in optimum culture conditions. Thereafter, media in each well was eliminated and wells were reinstated with 10 μM DCFH-DA followed by another incubation of 30 min in dark. Eventually, the cells were analyzed for their DCF-DA-mediated fluorescence intensity through Synergy H1 Hybrid Reader, BioTek, Winooski, VT, USA fluorescent microplate reader at an excitation/emission wavelength of 485/528 nm. The fluorescent intensity was expressed as average fluorescence percent (%) in comparison with untreated control.

2.2.4. Assessment of AsA Induced Apoptosis Using Acridine Orange (AO) Staining

Initiation of apoptosis within PC-3 cells exposed to AsA was estimated with AO and EtBr double staining. The apoptotic PC-3 cells were visualized using a green filter of Floid Imaging station as per the adopted protocol with subtle modifications [23]. Briefly, $1 \times 10^5$ PC-3 cells were exposed to the above-stated concentrations of AsA in a 6-well plate for 24 h under standard culture conditions. Cells were subsequently detached through gentle scarping and centrifuged (1500 rpm; 2 min at 4 °C), and the pellet was subjected to gentle washing using cold PBS. Thereafter, a 10 μL of AO was used to treat the pellets for 15 min. The suspension was then visualized on a glass slide.

2.2.5. Evaluating Caspase-3 Activity within AsA Exposed PC-3 Cells

Caspase-3 activity was assessed colorimetrically through commercially available kit within AsA-treated human prostate cancer PC-3 cells by adhering to the manufacturer's

instruction. Briefly, $3 \times 10^6$ PC-3 cells ensuing AsA treatment at stated concentrations were lysed for 10 min using 50 μL of ice cold lysis buffer placed on ice. The suspension obtained thereafter was subjected to centrifugation for 1 min at 10,000 rpm and 4 °C after which the supernatant was extracted and placed again on ice. Subsequently, 50 μL of lysate was placed in each well of a 96-well format and mixed with equi-volume of 10 mm DTT. Finally, 4 mM DEVD-pNA was supplemented further to each well and allowed to react for 10 min at RT. The plate was then assessed for the absorbance of each well at 405 nm through a microplate reader Bio-Rad, Hercules, CA, USA. Alteration within caspase activity was substantiated in percent (%) in correlation with untreated PC-3 control cells.

### 2.2.6. Assessment of Caspase-3 Inhibitor Mediated Effects on Human Prostate Cancer PC-3 Cells

AsA-induced toxic effects on human prostate cancer PC-3 cells was further delineated through capase-3 inhibitor Z-DEVD-FMK. Primarily, PC-3 cells were pretreated using Z-DEVD-FMK (50 μM; 2 h). Subsequently, cells were exposed to AsA (stated concentrations) and left undisturbed for 24 h under standard culture conditions. Eventually, the viability of AsA-treated PC-3 cells was assessed through standard MTT as outlined in Section 2.2.2.

### 2.3. Statistical Analysis

Experimental data reported here are mean ± SEM of three discrete experiments performed thrice. Statistical analysis was determined using GraphPad Prism Ver.5.0, San Diego, CA, USA to ascertain significance levels when the value of probability was <0.05 between different groups through student paired t-test, one way-Anova, subsequently followed by Dunnett post-hoc test. * $p < 0.05$, ** $p < 0.01$ and *** $p < 0.001$.

## 3. Results

### 3.1. AsA Reduced the Cell Viability of PC-3 Cells N

To investigate the inhibitory effects of AsA on PCa cells, MTT assay was performed. The cells were treated with different concentrations of AsA (0–80 μM) for 24 h. As shown in Figure 1A, it was found that AsA substantially inhibited the growth of PC-3 cells by 77.96 ± 2.98% (20 μM; $p < 0.05$), 58.71 ± 4.83% (40 μM; $p < 0.01$), and 26.85 ± 5.10% (80 μM; $p < 0.001$) in a dose-dependent manner.

### 3.2. AsA Instigated Apoptotic Cell Death within PC-3 Cells

To investigate the apoptotic cell death in AsA-treated PC-3 cells, cells were counted to investigate the number of viable cells (VI) and apoptotic cells (AO) by using fluorescence microscope. As observed in phase contrast micrographs in Figure 1B, apoptotic cells emitted diffuse green fluorescence by using AO within the fragmented DNA of AsA-treated PC-3 cells. Moreover, untreated control cells were distinguished by the presence of bright green fluorescence due to the presence of intact nuclei within the cells. At the highest dose (80 μM) of AsA, nuclear chromatin condensation and blebbing were observed, which are considered to be peculiar attributes of apoptosis. The results showed that AsA induced morphological aberrations in the nucleus following a dose-dependent trend, which ultimately was associated with apoptosis.

Furthermore, apoptotic cell death was quantitatively assessed in AsA-treated PC-3 cells. As shown in Figure 1C, dose-related decrease of viable cells (VI) with diffused green fluorescence was 45.61 ± 4.97% (20 μM), 28.92 ± 4.83% (40 μM), and 11.56 ± 5.10% (80 μM) as compared to control, where the number of VI was 75.31 ± 4.87. Contrary to this, AO cells increased to 15.69 ± 0.39% (20 μM), 19.33 ± 0.48% (40 μM), and 33.65 ± 0.45% (80 μM) as compared to the control (8.56 ± 0.59%). Thus, our results signified that AsA significantly induced apoptotic cell death in PC-3 cells.

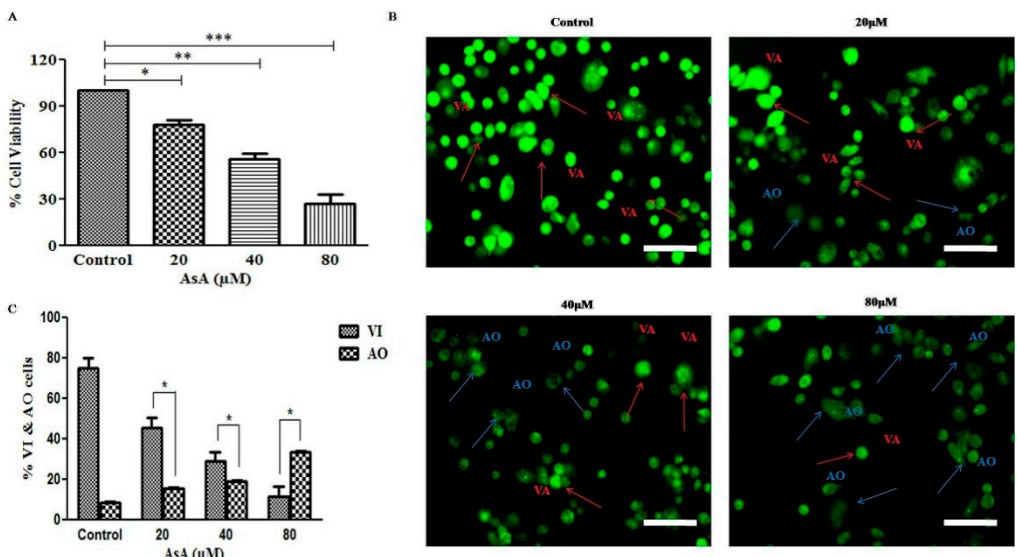

**Figure 1.** (**A**) Cell viability percentage (%) of human androgen-independent PC-3 cells exposed to various concentrations (20, 40, and 80 µM) of AsA; (**B**) photomicrographs stained with acridine orange, indicating the viable (VI) and apoptotic (AO) PC-3 cells post treatment with AsA (as indicated by the arrows; scale bar = 100 µm); and (**C**) quantification of both the VI and AO PC-3 cells. Experimental data reported here are mean ± SEM of three discrete experiments performed thrice. Statistical significance between control and treated groups was analyzed using student paired t-test, one-way Anova, and Dunnett post-hoc test as per the applicability where significance was illustrated when * $p < 0.05$; ** $p < 0.01$; and *** $p < 0.001$.

### 3.3. AsA Augmented the Levels of ROS in PC-3 Cells

ROS generation was found to be associated with the apoptosis induction in cancer cells [24]. Therefore, it is necessary to determine the level of ROS in PC-3 PCa cells after treatment with various doses (20–80 µM) of AsA for 24 h. As shown in Figure 2A, substantial increase in the intracellular level of ROS was enhanced by $56.71 \pm 3.04\%$ ($p < 0.001$) compared to untreated cells, following treatment with the 20 µM dose of AsA. Indeed, ROS generation was further enhanced to $116.01 \pm 4.18\%$ and $191.75 \pm 4.55\%$ ($p < 0.001$) in PC-3 cells at the concentrations of 40 µM and 80 µM AsA, respectively. These results suggested that the treatment of AsA augments the ROS generation in PCa cells.

### 3.4. Assessment of Caspase-3 Activity in AsA-Treated PC-3 Cells

To investigate the mechanism underlying AsA-mediated apoptosis in PC-3 cells, we inspected the intracellular caspases-3 activity in AsA-treated PC-3 cells. As observed in Figure 2B, our findings suggested a substantial increase in the caspase-3 activity to $26.72 \pm 3.94\%$, $59.05 \pm 4.51\%$, and $124.18 \pm 4.95\%$ ($p < 0.05$, $p < 0.001$) as compared to control, at the indicated doses of 20, 40, and 80 µM of AsA, respectively. Thus, our results further confirm that the antiproliferative efficacy of AsA was mediated by inducing apoptosis in PC-3 cells.

### 3.5. Attenuation of AsA-Mediated Apoptosis in PC-3 PCa Cells by Caspase Inhibitors

To characterize whether the caspases are involved in the AsA-mediated cytotoxicity in PCa cells, PC-3 cells were initially treated with 50 mm caspase-3 inhibitor (Z-DEVD-FMK) for 2 h and then subsequently treated with AsA at the indicated doses for 24 h. Further, MTT assay was used to determine the cell viability as described above. Pretreatment with capase-3 inhibitor substantially decreased the amount of cytotoxicity in PCa cells caused by the treatment of AsA Figure 2C. These findings suggested that the induction of caspase-3 activity played a critical role in AsA-mediated apoptosis.

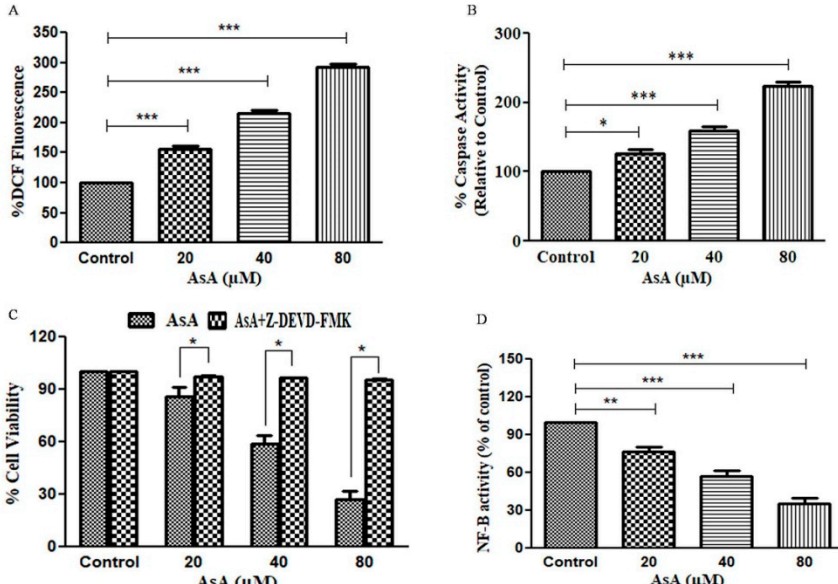

**Figure 2.** (**A**) Quantitative evaluation of AsA-mediated DCF-DA fluorescence intensity (%) within PC-3 cells; (**B**) AsA-mediated instigation of caspase-3 at varying concentrations of 20, 40, and 80 μM in PC-3 cells; (**C**) cell viability (%) of PC-3 cells pretreated with caspase-3 inhibitors and (**D**) percent activity of NF-κB transcription factor. Experimental data reported here are mean ± SEM of three discrete experiments performed thrice. Statistical significance between control and treated groups was analyzed using student paired *t*-test, one-way Anova, and Dunnett post-hoc test as per the applicability where significance was illustrated when * $p < 0.05$; ** $p < 0.01$ and *** $p < 0.001$.

### 3.6. AsA Inhibits the Activation of NF-κB in PC-3 Cells

The NF-κB signaling pathway plays a critical role in the survival of cancer cells and is widely known for regulating the expression of Bcl-2 proteins [25]. Thus, we assessed the NF-κB activity in PC-3 cells to investigate the efficacy of AsA on NF-κB activity. AsA treatment led to a dose-dependent decrease in NF-κB activity to 76.53 ± 3.37%, 57.20 ± 3.78% ($p < 0.01$) and 35.42 ± 3.37% ($p < 0.001$) as shown in Figure 2D, indicating that AsA mediated the inhibition of NF-κB in PCa cells.

### 4. Discussion

Despite the escalation within incidences and mortality rates of PCa, available therapeutic modalities are not substantially effective in prolonging the mean survival time period of patients. Therefore, there is a quest for exploration and development of novel therapeutic agents that could exhibit enhanced therapeutical efficacy with concomitantly reduced side-effects against PCa patients. Occurrence of tumors involves the evasion of apoptosis, resulting in an uncontrolled proliferation of tumor cells. Thus, an efficient cancer therapy may focus towards utilizing the cytotoxic potential of natural compounds in destroying the malignant cells by triggering the apoptotic pathways with least side effects [26]. The results from our investigation revealed that AsA may exert chemopreventive potential against PCa by modulating certain key cell signaling molecule. It was found that the triterpenoids (AsA) exerted substantial anti-proliferative effects on PC-3 cells via inducing caspase-mediated apoptosis along with suppressing the stimulation of proinflammatory transcription factor NF-κB, which is chiefly involved in tumor survival and proliferation.

Apoptosis or programmed cell death is an extremely controlled process that plays an imperative role in cell death and is also indispensable in several cellular functions from fetal development to adult tissue homeostasis [27]. Available literature clearly outlines that apoptosis results due to multiple biochemical changes in cells, which primarily includes nuclear condensation and fragmentation, alteration in the mitochondrial membrane

potential, and regulation of caspases [28]. In our present investigation, we explored the cytotoxic potential of AsA on androgen-independent PC-3 cell line. The cell viability analysis suggested that AsA significantly suppressed the proliferation of these cells. The results substantiated that the exposure of AsA strongly curtailed the viability of PC-3, where the reduced viability was directly correlated with the concentration of AsA. Since the exposure of AsA exhibited a strong cytotoxic potential on PC-3 cells, we performed AO fluorescent dye staining to explore the cells undergoing apoptosis and formation of apoptotic bodies within the PCa cells. Although AO photomicrographs clearly demonstrated the morphological alterations associated with apoptosis, we also quantified the population of apoptotic cells by using Image J software (Ver. 1.46r). The observations compelled us to conclude that AsA instigated a substantial dose-dependent increase in apoptosis in PCa.

Previously reports have documented that oxidative stress serves to be a crucial impetus for altering mitochondrial membrane potential and apoptosis [29]. We measured the levels of ROS upon AsA treatment on PC-3 cells to inspect the involvement of ROS in the apoptotic pathways. The observations conclusively made it evident that there was a substantial dose-dependent augmentation within intracellular ROS within AsA-treated PC-3 cells. Caspases are considered to be paramount regulators of apoptosis [30]. Caspase-3 being a chief executioner of apoptosis is partially or completely accountable towards proteolytic cleavage of cellular proteins [31]. Therefore, we inspected the caspase activity in AsA-treated PC-3 cells in juxtaposition with the control and observed that AsA induced caspase- 3 activation, leading to apoptosis. Further, pretreatment with capase-3 inhibitor (Z-DEVD-FMK) significantly reduced AsA-induced cytotoxicity in PC-3 cells, implicating that stimulation of caspase-3 post-AsA exposure induced apoptosis.

Convincing evidence suggested that constitutive activation of NF-κB is a trademark of multiple human carcinomas [16,32]. Earlier researches have elucidated that NF-κB is constitutively over-expressed in androgen-independent DU145 and PC-3 PCa cells [33]. NF-κB is a protein complex involved in adjusting DNA transcription and is considered an apoptosis inhibitor [34]. Therefore, suppressing the functionality of NF-κB can induce apoptosis. During this investigation, we demonstrated that AsA portrayed its relevance in inhibiting the activation NF-κB in PC-3 cells. In line with these findings, we conclude that the apoptosis induced by AsA on PC-3 cells could plausibly be correlated with suppressed NF-κB activity. Conclusively, AsA could be a plausible therapeutic modality in the treatment of PCa.

## 5. Conclusions

On the basis of inferences drawn from this initial investigative study, AsA is competent in inducing apoptosis in PC-3 cells. The evidences presented herewith provide a connective link between anti-proliferation and apoptotic induction, and the cell death in PCa cells was a result of caspase activation. These findings implicated that apoptosis occurs through intrinsic apoptotic signaling pathways with regulation of NF-κB protein modulation. Thus, our results affirmed the potency of AsA as a chemotherapeutic agent in human PCa cells, and therefore, may be worthy for application in the optimization and development of novel therapeutical-based interventions against PCa. However, subsequent researches are further warranted to elucidate the molecular mechanics involved in efficacy of AsA as an anticancer agent against PCa cells.

**Author Contributions:** Conceptualization, A.A. (Ahmed Alafnan) and T.H.; methodology, S.M.D.R.; validation, S.M.D.R., A.M. and A.A. (Abdulwahab Alamri); formal analysis, A.A. (Ahmed Alafnan) and T.H.; resources, A.A. (Abdulwahab Alamri); writing—original draft preparation, A.M. and A.A. (Abdulwahab Alamri); writing—review and editing, S.M.D.R.; supervision, A.A. (Ahmed Alafnan). and T.H. All authors have read and agreed to the published version of the manuscript.

**Funding:** This research received no external funding.

**Institutional Review Board Statement:** Not applicable.

**Informed Consent Statement:** Not applicable.

**Data Availability Statement:** Not applicable.

**Acknowledgments:** The authors acknowledge the support given by University of Hail, KSA.

**Conflicts of Interest:** The authors declare no conflict of interest.

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
