# Peer review of "Prostate Apoptotic Induction and NFκB Suppression by Dammarolic Acid: Mechanistic Insight into Onco-Therapeutic Action of an Aglycone Asiaticoside"

_cimb, doi:10.3390/cimb43020066_

Round 1
Reviewer 1 Report
In this manuscript, Alafnan and collaborators investigated the cytotoxic capability of Dammarolic acid or Asiatic acid (AsA) against human prostate cancer cell line PC-3.
AsA is a bioactive compound, contained within several plants. Its pro-apoptotic effect was already explored in vitro in different types of human cancer cell lines, included cells of prostatic origin, PPC-1 (Gurfinkel DM., et al., 2006).
Despite similar experiments were performed on other cell lines, no previous studies reported the effect of AsA on PC-3 cells.
However, to support results, some additional information would be needed.
- Please provide information on the preparation of the AsA solution and the dilutions used in the experiments.
- In each experiments, were the “control” cells treated with same volume of diluent or nothing?
- In the legends of the figures it is reported “Experimental data reported here is mean ± SEM of three discrete experiments performed thrice”. How many experiments have been performed? Three? Or Three in triplicate?
- Cells viability in Figure 1A and Figure 1B is different. What is the difference between the two experiments?
- In Figure 1C, the comparison between VI and AO in each experimental condition is not informative. Authors should evaluate whether there is a trend in decreasing viability and increasing apoptosis in a dose-dependent manner.
Author Response
Acknowledgement of Reviewer’s comment and reply with reference to manuscript ID: cimb-1333035 submitted under Special Issue "Involvement of Medicinal Plants and Food in the Molecular Mechanisms of Disease Prevention".
Dear Honorable Reviewer,
We appreciate the kindness of the Reviewers in assisting us to improvise the quality of manuscript and providing sufficient time regarding the same during the current pandemic. All authors have reviewed and agreed to the submission of the revised manuscript. We hope that the manuscript is now acceptable for publication.
Reply to reviewer’s comments
Reviewer 1
Comment 1: Please provide information on the preparation of the AsA solution and the dilutions used in the experiments.
Reply: The author’s would like to bring in consideration of the learned reviewer that, AsA was weighed (1 mg) and was mixed by vortexing in 10 μl DMSO. Thereafter the volume was made upto 1ml in complete culture media (with only 5% serum concentration). Eventually from this stock of AsA (1 mg/ml; 2046.24 μM), respective dilutions namely 20, 40 and 80 μM working stocks were prepared using N1V1=N2V2
Comment 2: In each experiments, were the “control” cells treated with same volume of diluent or nothing?
Reply: Authors would like to state that during the experimentation, the control cells only received the culture media with a serum concentration of only 5%.
Comment 3: In the legends of the figures it is reported “Experimental data reported here is mean ± SEM of three discrete experiments performed thrice”. How many experiments have been performed? Three? Or Three in triplicate?
Reply: The authors would like to bring in consideration of the learned reviewer that each experiment was performed in triplicate thrice.
Comment 4: Cells viability in Figure 1A and Figure 1B is different. What is the difference between the two experiments?
Reply: Authors extend their apologies for this avoidable typological error in the graph of Fig. 1C. In accordance with the learned reviewer’s recommendation, authors would like to state that Fig. 1A represents the cell viability whereas Fig. 1C represents percentage of viable (VI) and apoptotic cells (AO). Nevertheless, the required changes have been incorporated and included within the main manuscript.
Comment 5: In Figure 1C, the comparison between VI and AO in each experimental condition is not informative. Authors should evaluate whether there is a trend in decreasing viability and increasing apoptosis in a dose-dependent manner.
Reply: Authors would like to state that in Fig. 1C, the comparison between the VI and AO in each condition was a preliminary assay. However, in accordance with the learned reviewer comments, quantitative assays suggesting the apoptosis induction will be included in coming articles to further substantiate a dose-dependent anticancer manner.
Reviewer 2 Report
The triterpenoid compound asiatic acid (AsA) is a plant derivate from the tropical medicinal plant Centella asiatica. Previous studies suggested that AsA possessed a wide range of pharmacological properties, including antitumor activities. As for antitumor effect, AsA was reported to induce apoptosis in hepatic, gastric, breast cancer cell lines and other tumor cells. One of the potential mechanism is as follows: AsA induces caspase-9 activity, which further activates caspase-3, resulting in irreversible apoptotic death in the tumor cells (DOI: 10.1248/bpb.32.1399). In the paper by Gurfinkel et al. PC-1 prostate cancer cells were treated with AsA: AsA induced rapid caspase-dependent and independent cell death, while AsA-induced death was associated with early activation of caspases 2, 3, and 8, but not caspase 9 ( DOI: 10.1007/s10495-006-9086-z).
Here the authors found that AsA exerted substantial anti-proliferative effects on PC-3 cells via inducing caspase-mediated apoptosis along with suppressing the stimulation of proinflammatory transcription factor NF-κB.Furthermore, the effects were dose-dependent. The experiments were clearly described and the methodology is adequate.
There are several typing errors to be corrected, together with punctuation (e.g. lines 57, 65, 70, 111, 113, 177: ',which'; line 89 'n'; line 109 ' cellnns'.).
Please kindly rewrite lines 44-46 as these management is too confusing for the reader.
Could you desrcibe why the resepctive dosages of AsA were chosen?
Author Response
Acknowledgement of Reviewer’s comment and reply with reference to manuscript ID: cimb-1333035 submitted under Special Issue "Involvement of Medicinal Plants and Food in the Molecular Mechanisms of Disease Prevention".
Dear Honorable Reviewer,
We appreciate the kindness of the Reviewers in assisting us to improvise the quality of manuscript and providing sufficient time regarding the same during the current pandemic. All authors have reviewed and agreed to the submission of the revised manuscript. We hope that the manuscript is now acceptable for publication.
Reply to reviewer’s comments
Reviewer 2
Comment 1: There are several typing errors to be corrected, together with punctuation (e.g. lines 57, 65, 70, 111, 113, 177: ',which'; line 89 'n'; line 109 ' cellnns'.).
Reply: Authors extend their apologies for these avoidable typological errors. In accordance with the learned reviewer’s recommendation, the required changes have been incorporated and highlighted within the main manuscript.
Comment 2: Please kindly rewrite lines 44-46 as these management is too confusing for the reader.
Reply: As per the recommendations of learned reviewer the aforesaid line has been rewritten and included within the main text as “Standard therapeutical management of these malignancies includes surgical, chemo/radio-therapeutics and hormonal or immunotherapeutics.”
Comment 3: Could you describe why the respective dosages of AsA were chosen?
Reply: Authors would like to state that at the time of experimentation, a range of concentrations of AsA were undertaken; however, out of them only these concentrations (20, 40 and 80 µM) were screened out as the effect of AsA was more significant at these concentrations.